# Exercise Promotion System for Single Households Based on Agent-Oriented IoT Architecture

**DOI:** 10.3390/s24072029

**Published:** 2024-03-22

**Authors:** Taku Yamazaki, Tianyu Fan, Takumi Miyoshi

**Affiliations:** College of Systems Engineering and Science, Shibaura Institute of Technology, Saitama 337-8570, Japan; bp18097@shibaura-it.ac.jp (T.F.); miyoshi@shibaura-it.ac.jp (T.M.)

**Keywords:** single household, exercise promotion, internet of things, agent-oriented architecture

## Abstract

People living alone encounter well-being challenges due to unnoticed personal situations. Thus, it is essential to monitor their activities and encourage them to adopt healthy lifestyle habits without imposing a mental burden, aiming to enhance their overall well-being. To realize such a support system, its components should be simple and loosely coupled to handle various internet of things (IoT)-based smart home applications. In this study, we propose an exercise promotion system for individuals living alone to encourage them to adopt good lifestyle habits. The system comprises autonomous IoT devices as agents and is realized using an agent-oriented IoT architecture. It estimates user activity via sensors and offers exercise advice based on recognized conditions, surroundings, and preferences. The proposed system accepts user feedback to improve status estimation accuracy and offers better advice. The proposed system was evaluated from three perspectives through experiments with subjects. Initially, we demonstrated the system’s operation through agent cooperation. Then, we showed it adapts to user preferences within two weeks. Third, the users expressed satisfaction with the detection accuracy regarding their stay-at-home status and the relevance of the advice provided. They were also motivated to engage in exercise based on a subjective evaluation, as indicated by preliminary results.

## 1. Introduction

Single households have become common worldwide for various reasons [1]. Specifically, Japan is experiencing an increase in single-person households, alongside a declining birthrate and an aging population. A national survey across Japan [2] reported the number of single-person households in Japan to be approximately 1491 million, corresponding to 28.8% of all households; this is the largest number and percentage of single-person households reported since the survey began. Another survey [3] reported the percentages of single-person households to be 39.3% of all households and the number of single households with inhabitants aged 65 and above to be 22.9% because of the increasing number of unmarried individuals. With this increasing number of single-person households, ensuring the well-being of their inhabitants in future societies is important. However, inhabitants of single-person households encounter challenges in achieving well-being due to a lack of awareness about their situations and lifestyles, leaving them without the necessary support. For example, they often do not have anyone to take care for them during injuries or sickness [4]. Furthermore, their efforts to manage health effectively, like maintaining a healthy diet, getting sufficient sleep and exercise, and having orderly lifestyles, are hindered by a lack of awareness about their lifestyles [5]. Recently, the surge in remote working due to the COVID-19 pandemic has led many people to abandon their exercise routines [6]. A previous study [7] confirmed the physical benefits of promoting exercise, even when users viewed only an on-demand physical activity video.

Therefore, an easy and continuous support system that assists single-person households in passively practicing good health behaviors and acquiring good lifestyle habits must be developed [8]. To support our daily lives efficiently, smart homes based on the internet of things (IoT), which realizes smart sensing, monitoring, and automation using various IoT devices, have been studied, developed, and widely adopted [9,10,11]. Considering the health dynamics in single-person households, support systems need to accurately assess their activities and consistently encourage residents to adopt healthier behaviors without causing mental strain. From the viewpoint of system components, support system functions must be simple and loosely coupled to handle various smart home applications. Agent-oriented IoT [12,13,14], a distributed approach for designing IoT systems [15,16], realizes various services [17,18,19,20] based on autonomous and flexible cooperation among agents implemented on devices [21].

Thus, in this study, we propose an exercise promotion system for single-person households to help individuals living alone adopt good lifestyle habits. The system comprises autonomous IoT devices as agents and is based on the agent-based IoT architecture. The proposed system estimates user activity via sensors and offers advice regarding exercises to the user based on their recognized conditions, surroundings, and preferences. We designed each agent and its algorithms in the proposed system and implemented them on embedded devices and smartphones. Finally, the proposed system was evaluated by installing it in the homes of some participants. The main contributions of this study are summarized as follows.
We extracted fundamental functions from IoT smart home elements and modeled them as simplified IoT agents playing roles based on periodic tasks and request–response-based behaviors. The proposed model enhances the flexibility of the system composition and extensibility for deploying divergent services in various home environments. Additionally, we present an example of the design and implementation of an exercise promotion system based on autonomous cooperation among these simple IoT agents.We incorporated a user feedback mechanism into the exercise promotion system to adapt the advice to the user’s preference. Therefore, the proposed system can generate appropriate exercise recommendations to promote and motivate users based on their situations, surroundings, and exercise preferences. Additionally, it enhances its advice content based on user feedback.We confirmed that the proposed system can be implemented by autonomous cooperation among simplified IoT agents through actual experiments on each subject’s home from the perspective of IoT system design. Additionally, we confirmed that the proposed system operates properly in various home environments to validate the behavior of the entire proposed system in experiments. Therefore, we proved that the proposed system is flexible and can be deployed in diverse home environments.We revealed the impact of the adaptation algorithm on the content of the advice through experiments. Experimental results confirmed that the proposed system can appropriately adjust the weights of keywords related to the probabilities of keyword selection to generate advice within one or two weeks in most cases. Finally, the proposed system was subjectively evaluated through preliminary experiments by implementing it in each subject’s home from the comprehensive perspective of exercise promotion based on advice. The subjective evaluation suggested that the proposed system can motivate individuals to exercise by offering timely appropriate advice based on the estimated subject’s status and keywords adapted to the subject’s preference based on the user feedback.

The remainder of this paper is organized as follows. Section 2 introduces related work from the viewpoints of activity recognition and IoT-based healthcare. Section 3 describes the design and implementation of the proposed exercise support system based on agent-based IoT. Section 4 evaluates the proposed system via three experiments. Finally, Section 5 concludes the study.

## 2. Related Work

Recently, numerous studies have been conducted to support daily lives [22,23,24,25,26,27,28,29,30,31,32]. IoT-based smart home appliances can perform various functions, including home surveillance, monitoring, and automation, by linking home equipment and smartphones via cloud services. Similar to this study, an IoT smart home system for supporting the lives of elderly individuals [22], an IoT framework for elderly people [23], and health monitoring using a wristband device [24] were developed. However, many existing studies for supporting independent living, including the above studies, have primarily focused on achieving health and lifestyle monitoring of elderly people to detect anomalies [25]. Moreover, monitoring individuals at home without relying on wearable devices is crucial, given the varying receptivity towards such devices among individuals and the inherent inconvenience of wearing them continuously.

Various status recognition algorithms and their systems have been proposed from the perspective of status recognition of individuals in healthcare [26,27,28,29]. They primarily focused on the status recognition of users based on collected sensor information. However, to analyze the effectiveness of IoT-based healthcare, the overall system must be evaluated depending on the use-cases, as the actual effects of support systems depend on various factors, in addition to recognition accuracy.

Various activity recognition systems using IoT devices have been proposed from the perspective of activity recognition [30,31,32]. However, they primarily focused on the recognition technique and its performance in a smart home environment. Therefore, to achieve behavior change in single households, support mechanisms for their lives and motivation are required for real systems.

In addition, from the viewpoint of system design, commercial IoT systems provide simple command sets to realize services by ensuring cooperation between devices through cloud services [33,34,35]. However, if the cooperation for a service becomes complex and tightly coupled, such as a cloud-based approach, managing the services becomes challenging, and vendor lock-in may occur. In addition, although a monitoring architecture based on fog IoT [36] was proposed for effective cooperation among sensors, the implementation design of sensor nodes has not been discussed. Therefore, the design concept should be defined to simplify device behaviors. For example, each device autonomously provides simple functions as an agent and is loosely coupled to reduce the complexity of the system design.

## 3. Exercise Promotion System for Single Households

### 3.1. Overview

Herein, we propose an exercise promotion system for single households using IoT devices in a discipline of agent-oriented IoT architecture to promote the development of good lifestyle habits without imposing a mental burden.

In the proposed system, each IoT device, as an agent, provides periodic tasks and simple request-and-response-based functions for playing a specific role. Their roles and behaviors are clearly defined, and the exercise promotion service is implemented through autonomous cooperation among agents without relying on a centralized server. This agent-oriented approach offers extensibility and flexibility for deployment in diverse cases and for various services. Additionally, the proposed system passively provides users with support at an appropriate time by estimating their activity. It then offers advice to users based on their preferences. Finally, it adapts the contents of the advice based on user feedback to reduce the mental burden, motivating the user to exercise.

The proposed system comprises a user smartphone and four agents: (1) sensing agent, (2) activity recognition agent, (3) advice agent, and (4) feedback agent. The sensing agent detects whether the user is at home using sensors and determines whether the user has been at home for an extended period. The activity recognition agent estimates the user’s current activity using a camera to check for activities such as resting and working. The feedback agent obtains feedback from the user regarding exercise advice and activity recognition. In this study, only a smartphone was used to receive and read advice from the advice agent. Figure 1 illustrates the relationships and interactions between the user, agents, and user’s smartphone in the proposed system. Section 3.2 describes agent designs and behaviors in detail.

Figure 2 illustrates an example of the workflow of the proposed system when the sensing agent determines that the user has remained at home for a long duration. First, the sensing agent periodically attempts to detect whether the user remains at home. If the sensing agent determines that the user has remained at home for a long time, it sends a notification to the advice agent. When the advice agent receives a notification, it sends a request to recognize the user’s current activity status to the activity recognition agent. The activity recognition agent then captures a photo, estimates the user status from the captured photo using a machine learning algorithm, and returns the recognized user status to the advice agent. If the user is recognized as resting, the advice agent generates advice with the method and location of the exercise based on both weather information and the user preference for exercise collected from the information searched on the internet and sends the advice to the user’s smartphone. Subsequently, the advice agent requests feedback on the advice for the feedback agent. The user provides feedback to the feedback agent, which sends feedback to the advice agent. Consequently, the advice agent updates the user preference based on feedback to adapt the contents of the advice and make it more suitable for the user. Additionally, the advice agent requests feedback on the status recognized by the user for the feedback agent. The user provides feedback on the recognized status (e.g., resting or working) to the feedback agent, and the feedback agent then sends feedback to the advice agent. Subsequently, the advice agent relays the result to the activity recognition agent, which records the pair of captured photo and feedback as the ground-truth label. The activity recognition agent periodically updates the estimation model based on the stored training dataset to improve the recognition accuracy. Thus, the proposed system can generate appropriate advice based on user conditions, surroundings, and preferences to promote exercise.

### 3.2. Specifications of Agents

This section describes the detailed designs and algorithms for each agent using pseudocodes. Table A1 lists the notations used to express the proposed system and the parameter values used in the experiments, as presented in the Appendix A.

#### 3.2.1. Sensing Agent

The sensing agent ${\mathrm{AGT}}_{\mathrm{sense}}$ recognizes whether the user is at home or has left. Algorithm 1 presents an algorithm for sensing agents. First, the agent attempts to detect whether the user stays at home with an interval $T_{\mathrm{int}}^{\mathrm{attempt}}$ using a detection sensor, and counts the number of detections $n_{\mathrm{detect}}$ until a certain number of attempts $N_{\mathrm{attempt}}^{\mathrm{th}}$ as a period. If $n_{\mathrm{detect}}$ is similar to or larger than the threshold $N_{\mathrm{detect}}^{\mathrm{th}}$, the agent recognizes the user as remaining at home until that period and counts the number of periods the user remained at home $n_{\mathrm{remain}}$. If $n_{\mathrm{remain}}$ reaches $N_{\mathrm{remain}}^{\mathrm{th}}$, the agent recognizes that the user has remained at home for an extended period and sends a notification to the advice agent.

#### 3.2.2. Advice Agent

The advice agent ${\mathrm{AGT}}_{\mathrm{adv}}$ generates advice for exercise upon receiving a request from other agents. Algorithm 2 presents an algorithm for advice agents. The advice comprises the exercise method, location, and weather information. The exercise method and its location are determined based on the user’s location and preference.

To estimate user preference and suggest an appropriate method for each user, the agent has an item set M, which denotes the vector of items including the objective, method, and intensity of the exercise to determine the content of the exercise, defined as $M=\{S_{i}|i\in Z^{+}\}$. Here, the *i*-th item $S_{i}$ contains the vector of pairs of a keyword (e.g., work out, boxercise, and yoga, in the case of exercise method) and weight, and are defined as $S_{i}=\left\{s_{i,j}|j\in Z^{+}\right\}$, where the *j*-th pair in the *i*-th item $s_{i,j}=\left\{k_{i,j},w_{i,j}\right\}$ comprises keyword $k_{i,j}$ and weight $w_{i,j}$.

To generate advice, this agent selects a keyword from each item $S_{i}$ in the item set M based on the keyword selection algorithm described in Algorithm 3. This algorithm randomly selects the *j*-th pair $s_{i,j}$ from an item $S_{i}$ using the suggestion probability $p_{i,j}$ calculated by:(1)$$p_{i,j}=\frac{w_{i,j}\in s_{i,j}}{\sum_{s\in S_{i}}w\in s},$$
based on the proportion of its weight $w_{i,j}\in s_{i,j}$ to the sum of the weights of all the keywords in the item. This agent generates a keyword set $K_{r}$ with selected keywords in the *r*-th round and generates advice to search for information from the internet using the keyword set $K_{r}$. Next, the advice agent sends the generated advice to the user’s device, such as a smartphone, and requests feedback from the user through the feedback agent. When the agent receives feedback for advice, it updates the weights of the keywords based on whether the feedback is positive or negative using a weight update algorithm, as described in Algorithm 4. If it receives positive feedback, it increases or decreases the weight of the keyword by $w_{\mathrm{up}}$ or $w_{\mathrm{down}}$ no more than $w_{\max}$ and no less than $w_{\min}$, based on whether the keyword is included in the *r*-th round keyword set $K_{r}$. With negative feedback, it decreases the weights of keywords $K_{r}$ in the *r*-th round to generate advice by $w_{\mathrm{down}}$, but no less than $w_{\min}$. Thus, the advice agent generates more suitable advice for users by updating the weights of the keywords for each item based on user feedback.
**Algorithm 1** Sensing agent ${\mathrm{AGT}}_{\mathrm{sense}}$.1:**function** initialize2:    Time interval of attempts $T_{\mathrm{int}}^{\mathrm{attempt}}\leftarrow \mathrm{any}\mathrm{time}\mathrm{interval}\mathrm{of}\mathrm{attempts}$3:    Current number of attempts in a period $n_{\mathrm{attempt}}\leftarrow 0$4:    Current number of detections in a period $n_{\mathrm{detect}}\leftarrow 0$5:    Current number of periods of remaining at home $n_{\mathrm{remain}}\leftarrow 0$6:    $N_{\mathrm{attempt}}^{\mathrm{th}}\leftarrow \mathrm{number}\mathrm{of}\mathrm{attempts}\mathrm{to}\mathrm{detect}\mathrm{whether}\mathrm{the}\mathrm{user}\mathrm{stays}\mathrm{at}\mathrm{home}$7:    $N_{\mathrm{detect}}^{\mathrm{th}}\leftarrow \mathrm{threshold}\mathrm{to}\mathrm{recognize}\mathrm{the}\mathrm{user}\mathrm{has}\mathrm{remained}\mathrm{at}\mathrm{home}\mathrm{for}a\mathrm{period}$8:    $N_{\mathrm{remain}}^{\mathrm{th}}\leftarrow \mathrm{threshold}\mathrm{to}\mathrm{recognize}\mathrm{the}\mathrm{user}\mathrm{has}\mathrm{remained}\mathrm{at}\mathrm{home}\mathrm{for}a\mathrm{long}\mathrm{time}$9:    setTimer($T_{\mathrm{int}}^{\mathrm{attempt}}$, self.timerExpired())10:**end function**11: 12:**function** timerExpired13:    $n_{\mathrm{attempt}}\leftarrow n_{\mathrm{attempt}}+1$14:    e←self.detectByPIRSensor()15:    **if** e=STATE_STAYING **then**16:        $n_{\mathrm{detect}}\leftarrow n_{\mathrm{detect}}+1$17:    **end if**18:    19:    **if** $n_{\mathrm{round}}=N_{\mathrm{attempt}}^{\mathrm{th}}$ **then**20:        **if** $n_{\mathrm{detect}}\geq N_{\mathrm{detect}}^{\mathrm{th}}$ **then**21:           $n_{\mathrm{remain}}\leftarrow n_{\mathrm{remain}}+1$22:        **end if**23:        $n_{\mathrm{attempt}}\leftarrow 0$24:        $n_{\mathrm{detect}}\leftarrow 0$25:    **end if**26:    27:    **if** $N_{\mathrm{remain}}=N_{\mathrm{remain}}^{\mathrm{th}}$ **then**28:        **if** STATE_STAYING=self.detectbyIRSensor() **then**29:           invoke ${\mathrm{AGT}}_{\mathrm{advice}}$.startAdvice()30:           $n_{\mathrm{remain}}\leftarrow 0$31:        **end if**32:    **end if**33:    34:    self.setTimer($T_{\mathrm{int}}^{\mathrm{attempt}}$, self.timerExpired())35:**end function**
**Algorithm 2** Advice agent ${\mathrm{AGT}}_{\mathrm{adv}}$.1:**function** Initialization2:    Item set $M\leftarrow \{S_{1},S_{2},...,S_{i},...\}$3:    Unevaluated picked keyword sets K←⌀4:**end function**5: 6:**function** startAdvice7:    *r*-th estimation result set $\{r,{\hat{y}}_{r}\}\leftarrow$ invoke ${\mathrm{AGT}}_{\mathrm{recog}}$.estimateState()8:    **if** ${\hat{y}}_{r}$ = STATE_REST **then**9:        call self.giveAdviceToUser()10:    **end if**11:    send information related to the estimation result ${\hat{y}}_{r}$ to the user’s smartphone12:    invoke ${\mathrm{AGT}}_{\mathrm{fb}}$.askResultOfStatusEstimation(*r*)13:**end function**14: 15:**function** giveAdviceToUser16:    *r*-th pair of the advice and picked keyword set $\left\{a_{r},K_{r}\right\}$    ← call self.generateAdvice()17:    $K\leftarrow K+K_{r}$18:    send *r*-th advice $a_{r}$ to the user’s smartphone19:    invoke ${\mathrm{AGT}}_{\mathrm{fb}}$.askDecisionForAdvice(*r*)20:**end function**21: 22:**function** generateAdvice(Round *r*)23:    *r*-th selected keyword set $K_{r}\leftarrow$self.generateKeywordSet(M)24:    generate *r*-th advice $a_{r}$ using $K_{r}$ based on the information from the internet25:    return the *r*-th pair of the advice and keyword set $\left\{a_{r},K_{r}\right\}$26:**end function**27: 28:**function** receiveFeedbackForDecision(Round *r*, Feedback $b_{r}$)29:    call self.updateWeights(M, $K_{r}\in K$, $b_{r}$)30:    $K\leftarrow K\setminus K_{r}$31:**end function**32: 33:**function** receiveFeedbackForStatusEstimation(Round *r*, Ground truth label $y_{r}$)34:    invoke ${\mathrm{AGT}}_{\mathrm{recog}}$.receiveLabel(*r*, $y_{r}$)35:**end function**
**Algorithm 3** Keyword selection algorithm.1:**function** generateKeywordSet(M)2:    Selected keyword set K←⌀3:    **for each** item $S_{i}\in$M **do**4:        $K\leftarrow K+k_{i,l}\in s_{i,l}$ selected from $S_{i}$ based on the probability $p_{i,l}$5:    **end for**6:    return K7:**end function**
**Algorithm 4** Weight update algorithm.1:**function** Initialization2:    Maximum or minimum weights $W_{\max}$, $W_{\min}$3:    Parameter to increase or decrease weights $W_{\mathrm{up}}$, $W_{\mathrm{down}}$4:**end function**5: 6:**function** updateWeights(Item set M, Selected keyword set K, feedback *b*)7:    **for each** item $S_{i}\in$M **do**8:        **for each** pair $s_{i,j}=\left\{k_{i,j},w_{i,j}\right\}\in$$S_{i}$ **do**9:           **for each** keyword $k_{l}\in$K **do**10:               **if** *b* = DECISION_ACCEPTED **then**11:                   **if** $k_{i,j}=k_{l}$ **then**12:                       $w_{i,j}\leftarrow$min($w_{i,j}+W_{\mathrm{up}},W_{\max}$)13:                   **else**14:                       $w_{i,j}\leftarrow$max($w_{i,j}-W_{\mathrm{down}},W_{\min}$)15:                   **end if**16:               **else if** *b* = REASON_DO_NOT_LIKE **then**17:                   **if** $k_{i,j}=k_{l}$ **then**18:                       $w_{i,j}\leftarrow$max($w_{i,j}-W_{\mathrm{down}},W_{\min}$)19:                   **end if**20:               **end if**21:           **end for**22:        **end for**23:    **end for**24:**end function**

#### 3.2.3. Activity Recognition Agent

The activity recognition agent ${\mathrm{AGT}}_{\mathrm{recog}}$ focuses on recognizing user activity status based on the requests of other agents. Algorithm 5 presents a detailed description of the proposed algorithm. Herein, this agent is assumed to recognize the user status based on a machine learning algorithm. When the system begins, the activity recognition agent trains a machine learning model $f_{\theta }$ using a training set $Z=\{z_{m}|m\in Z^{+}\}$, with an initial number of examples $N_{\mathrm{init}}^{\mathrm{examples}}$, where $z_{m}=\left\{x_{m},y_{m}\right\}$ denotes the *m*-th training example composed of a photo $x_{m}$ and its ground-truth label $y_{m}$. When the agent receives a request from other agents to recognize the user activity status, it captures a photo $x_{r}$ in round *r* and stores it in an unlabeled photo set X. The current round number *r* is determined based on the sum of the number of examples |Z| and the number of unlabeled examples |X| plus one. Subsequently, it recognizes the user’s current activity status ${\hat{x}}_{r}=f_{\theta }\left(x_{r}\right)$ for resting, working, and being absent from the captured photo $x_{r}$ using the trained model $f_{\theta }$. The agent sends the estimation result ${\hat{x}}_{r}$ in a round *r* to the requester. In parallel with the recognition function, it awaits feedback on the ground-truth label of the stored photos $y_{r}$. When it receives $y_{r}$ for the unlabeled photo $x_{r}$, it adds the training example $z_{r}=\left\{x_{r},y_{r}\right\}$ to the training set Z and removes photo $x_{r}$ from the unlabeled photo set X. The agent retrains the recognition model $f_{\theta }$ using the updated training set Z every $N_{\mathrm{int}}^{\mathrm{retrain}}$.

#### 3.2.4. Feedback Agent

The feedback agent ${\mathrm{AGT}}_{\mathrm{fb}}$ receives feedback from the user. Algorithm 6 presents a detailed procedure for the feedback agent. This agent displays multiple-choice questions and collects user responses regarding advice and the recognition of the user’s activity status. Moreover, it can present and inquire about the subsequent question based on the user’s selection.

When receiving a feedback request regarding the decision for advice $a_{r}$ in round *r*, the feedback agent presents various choices (e.g., accept, pending, or reject) and awaits a user response through its graphical user interface (GUI). If the user rejects the advice, the feedback agent presents choices for rejecting the advice (e.g., the user does not like the recommended advice, the advice is not useful information, or the exercise has been performed) and awaits a user response. Finally, the feedback agent sends user choices to the requester.

When receiving a feedback request for the recognition result of the current activity status using *r*-th captured photo, the feedback agent presents choices regarding the user’s current activity status (e.g., working, resting) and awaits the user’s response. Finally, it reports the user choice as the ground-truth label ($y_{r})$ for the *r*-th captured photo $x_{r}$.
**Algorithm 5** Activity recognition agent ${\mathrm{AGT}}_{\mathrm{recog}}$.1:**function** Iintialize2:    X←⌀3:    $Z\leftarrow \{z_{1},z_{2},...,z_{N_{\mathrm{init}}^{\mathrm{samples}}}\}$4:    Estimation model $f_{\theta }$ pre-trained by the training set Z5:    $N_{\mathrm{int}}^{\mathrm{retrain}}\leftarrow$ any retraining interval6:    $r\leftarrow N_{\mathrm{init}}^{\mathrm{samples}}+1$7:**end function**8: 9:**function** estimateState10:    capture *r*-th photo $x_{r}$ by its camera module11:    $X\leftarrow X+x_{r}$12:    *r*-th estimation result ${\hat{y}}_{r}\leftarrow f_{\theta }\left(x_{r}\right)$13:    send ${\hat{y}}_{r}$ to ${\mathrm{AGT}}_{\mathrm{adv}}$14:    r←r+115:**end function**16: 17:**function** receiveLabel($r,y_{r}$)18:    $Z\leftarrow z_{r}\in \left(x_{r},y_{r}\right)$19:    $X=X\setminus x_{r}$20:    **if** $|Z|\;\mathrm{mod}\;N_{\mathrm{int}}^{\mathrm{retrain}}=0$ **then**21:        update the model $f_{\theta }$ using the training set Z22:    **end if**23:**end function**

**Algorithm 6** Feedback agent ${\mathrm{AGT}}_{\mathrm{fb}}.$
1:**function** askDesicionForAdvice(Round *r*)2:    display a message to ask the decision for *r*-th advice3:    $b_{r}\leftarrow$ call self.waitForFeedbackFromUser()4:    **if** $b_{r}$ = DECISION_ACCEPTED **then**5:        invoke ${\mathrm{AGT}}_{\mathrm{adv}}$.receiveFeedbackForDecision($r,b_{r}$)6:    **else if** $b_{r}$ = DECISION_WITHHOLDING **then**7:        send a response to ${\mathrm{AGT}}_{\mathrm{adv}}$ //do nothing8:    **else if** $b_{r}$ = DECISION_REJECTED **then**9:        call self.askReasonForRejectionOfAdvice(*r*)10:    **end if**11:
**end function**
12: 13:**function** askReasonForRejectionOfAdvice(Round *r*)14:    display a message to ask the reason for rejection of *r*-th advice15:    $b_{r}\leftarrow$ call self.waitForFeedbackFromUser()16:    **if** $b_{r}$ = REASON_DO_NOT_LIKE **then**17:        invoke ${\mathrm{AGT}}_{\mathrm{adv}}$.receiveFeedbackForDecision($r,b_{r}$)18:    **else if** $b_{r}$ = REASON_NO_USEFUL_INFO **then**19:        invoke ${\mathrm{AGT}}_{\mathrm{adv}}$.giveAdvice()20:    **else if** $b_{r}$ = REASON_ALREADY_DONE **then**21:        send a response to ${\mathrm{AGT}}_{\mathrm{adv}}$ // do nothing22:    **end if**23:
**end function**
24: 25:**function** askResultOfStatusEstimation(Round *r*)26:    display a message to ask the success or failure of status estimation27:    $y_{r}\leftarrow$ call self.waitForFeedbackFromUser()28:    invoke ${\mathrm{AGT}}_{\mathrm{adv}}$.receiveFeedbackForStatusEstimation($r,y_{r}$)29:
**end function**



### 3.3. Implementation

We present the implementation design of the proposed system for actual experiments in home environments. All agents expose ports to communicate with one another through a local network. They send and receive information represented in a simple JSON format for autonomous cooperation among them. We utilized three types of embedded devices: M5Stack [37], Raspberry Pi [38], and NVIDIA Jetson Nano [39]. M5Stack, an Arduino-compatible embedded device, can handle simple tasks. Raspberry Pi and Jetson Nano, Linux-based embedded devices, can execute various complex tasks. Jetson Nano is well-suited for artificial intelligence (AI) tasks, enabling the smooth execution of machine learning algorithms using its GPU-based AI acceleration units. Our implementation serves as one example; thus users are free to choose any devices that meet the task requirements of agents.

#### 3.3.1. Implementation of a Sensing Agent

Herein, the sensing agent was implemented on two M5Stack [37] equipped with a passive infrared (PIR) sensor or infrared (IR) sensor to place them in appropriate positions. In the experiments, the sensing interval $T_{\mathrm{int}}$ was set to 6 min, number of attempts to assess whether the user remains at home for a period $C_{\mathrm{attempt}}^{\mathrm{th}}$ was set to 10 times, threshold to determine whether the user remains at home for a period $N_{\mathrm{detect}}^{\mathrm{th}}$ was set to five times, and threshold to determine whether the user remains at home for periods $N_{\mathrm{remain}}^{\mathrm{th}}$ was set to four times. Essentially, the sensing agent detects whether the user stays at home, with 1 h as the period. If it detects that the user has remained at home for 4 h, that is, four periods, the sensing agent sends a notification to the advice agent about remaining at home for a long time. In our experiments, this agent was activated from 8:00 to 18:00, although the actual active time of each user depended on their lifestyles.

#### 3.3.2. Implementation of an Activity Recognition Agent

The activity recognition agent was implemented on NVIDIA Jetson Nano [39], a Linux-based embedded device equipped with an Sony IMX219 camera module, since the agent needs to execute a machine-learning-based classification task. This agent captures a photo using its camera module and estimates the user’s current activity status among resting, working, and being absent based on a pre-installed machine learning algorithm using the captured photo. Furthermore, GoogleNet (InceptionNet v1) [40] was employed to classify the activity status using a photo. The feedback agent utilizes 400 photos as $N_{\mathrm{init}}^{\mathrm{examples}}$, including situations of resting, working, and being absent, as the training set and creates the trained model $f_{\theta }$ using them.

#### 3.3.3. Implementation of an Advice Agent

The advice agent was implemented on Raspberry Pi 3B+ [38], a Linux-based embedded device, to obtain weather information related to the user’s current location, such as weather, temperature, pressure, humidity, and wind speed, using the Weather API from OpenWeatherMap [41]. Subsequently, the agent selects keywords $K_{r}$ in round *r* and gathers information based on the selected keywords $K_{r}$, weather information, and the places and locations for exercise using the Places API in the Google Maps Platform [42] within a certain range of user locations and the custom search JSON API on the Google Programmable Search Engine [43]. As keywords for exercise methods, we adopted the following methods: “workout”, “boxercise”, “yoga”, “walking”, “jogging”, “cycling”, “swimming”, “aqua aerobics”, “aerobics”, and “calisthenics”. As keywords for objectives for exercise, we adopted the following objectives: “lose weight”, “strength training”, “health maintenance”, “stress dissipation”, “dementia prevention”, and “break”. Finally, the agent generates an advice message based on this information and sends it to the user’s smartphone via a social messaging application. Herein, LINE [44], a common messaging application in Japan, was adopted to send notifications to the user; the LINE Notify API [45] sends a message from the device of the advice agent to the LINE application.

#### 3.3.4. Implementation of a Feedback Agent

The feedback agent was implemented on M5Stack, which has a display and push buttons to select display information, to obtain feedback for the system. Figure 3 illustrates the feedback flow in the current implementation of the proposed system. The agent displays the question-and-answer choices received from the other agent on its screen; subsequently, the subject chooses an answer by pushing a button. Subsequently, the feedback agent sends an answer to the requester to provide feedback.

## 4. Experiments

### 4.1. Experiment Setup

We conducted a three-day experiment with four subjects who lived alone in their homes where the proposed system was implemented. Table A1 lists the experimental parameters for the proposed system, as presented in the Appendix A. In the experiment, initial weights were assigned to the subjects based on their preferences to focus on the effectiveness of the advice. To assign weights to the subjects, they were first offered pseudo-advice and feedback based on their preferences to update the weights. Each subject repeated this step 20 times to obtain the initial values of the weights that reproduced approximately two weeks after running the proposed system. The subjects then recorded their history daily to analyze the experimental results while they spent their time.

Figure 4 illustrates the installation of the proposed system for a subject, although the arrangement depends on the subject. The sensing agents were positioned in locations frequented by the subjects; the activity recognition agent was set up in areas covered by cameras; the feedback agent was situated in easily accessible spots for the subjects to check and interact with the device; and the advice agent was installed in various locations. All the devices were connected to a wireless LAN router in their homes.

To evaluate the proposed system, the following evaluation indices were adopted: (1) the relationship between the actual sequence of the proposed system and the action history of the subjects, (2) time transitions of the weights of the subjects based on the rounds, and (3) subjective evaluation based on the questionnaire. Here, the relationship between the actual sequence of the proposed system and the action history of the subjects can be used to verify whether the proposed system works correctly and whether it realizes cooperation among agents during the actual subject activity. The time transitions of the weights of the subjects can be used to analyze actual changes in the weights along the subject choices. Subjective evaluation can uncover the effectiveness from the user’s perspective, focusing on the accuracy of detection and status estimation, the relevance of selected keywords for suggested advice, and user preferences. Thus, the proposed system can motivate subjects to exercise.

### 4.2. Experiment 1: Validation of System Sequence

Figure 5 and Figure 6 show the daily time sequences of the operation history of the proposed system and the action history from 10:00 to 22:00. Figure 5 shows the time sequences of the operation history of the proposed system from 18:03:10 to 18:04:10, when the proposed system detects that the subject was home for an extended period. First, Figure 5 indicates that the sensing agents periodically detect whether the subject remains at home using sensors. After waking up, the sensing agents correctly detected that the subject was staying at home, except when they left. When the sensing agent detects that the subject has remained at home for 4 h, it requests an estimation of the subject’s status from the activity recognition agent. Figure 6 indicates that the agents can cooperate autonomously; hence, the proposed system operates correctly. Therefore, the proposed system was implemented in each subject’s home to evaluate its effectiveness. Moreover, the proposed system is cost-effective because it is mainly composed of popular commodity IoT devices and smartphones with inexpensive general sensors.

These evaluation results indicate that the proposed system correctly runs on a user’s home with a single room. The proposed system can be deployed at a home with multiple rooms to put sensing agents in multiple points if necessary. The focus of the proposed system is to be implemented in single households; namely, the system is assumed to be implemented in a single or few rooms. However, the proposed system can adapt to a house with multiple rooms by deploying sufficient sensing agents in appropriate positions with a small modification in the cooperation sequence and their internal procedures. From a cost perspective, the proposed system is still reasonable if it is deployed at such a larger home, because sensing devices with IR and PIR sensors are inexpensive and easy to deploy at any position.

### 4.3. Experiment 2: Impact of User Feedback

Figure 7 and Figure 8 show the time transitions of the suggestion probabilities for the exercise types and objectives, respectively. The results indicated that probabilities changed to increase or decrease the weights based on the feedback rounds. In particular, the suggestion probabilities of keywords a subject prefers to become higher than the others based on the feedback given by the subjects via the implemented feedback agent.

We observed significant changes in probabilities from 10 to 20 rounds in some subjects. If a user regularly stays at home for a long time, the user receives advice approximately twice daily. In other words, the proposed system can proceed with two advice rounds daily, and therefore, the proposed system can adapt the user’s preferences to be appropriate for one to two weeks. However, the adaptation algorithm in the proposed system is a simple approach, with increasing and decreasing weights. Therefore, we can improve the algorithm by incorporating some state-of-the-art algorithms, including machine-learning-based approaches.

### 4.4. Experiment 3: Subjective Evaluations

Table 1 presents the subjective evaluation results of the proposed system after each subject experienced it for three days. The subjects were asked five questions about (1) the accuracy of detecting remaining at home for a long time, (2) the accuracy of estimating the subject’s status, (3) the appropriateness of advice corresponding to the selected keywords, (4) the appropriateness of advice corresponding to the subject’s preference, and (5) motivating the subjects to perform an exercise. Participants selected answers on a five-point scale from “strongly disagree”, “disagree”, “neutral”, “agree”, and “strongly agree”, with points assigned from one to five. Therefore, the neutral score is three points on the average.

Q1 denoted a subjective evaluation of the accuracy of detecting remaining at home for a long period. The subjects evaluated the detection accuracy highly because the proposed system precisely detected staying at home as shown in Figure 5. Q2 denotes the subjective evaluation of the accuracy of estimating the status of the subject. In contrast to Q1, the accuracy of the status estimation was evaluated to be close to the neutral score because a misunderstanding of the subject’s status might occur. Therefore, the subjects provided more feedback for status estimation through the feedback agent to retrain the model. Q3 and Q4 denote the appropriateness of the suggested advice for the selected keywords and subject preferences, respectively. These points were also highly valued by the participants. Therefore, the proposed system can precisely obtain information regarding exercises corresponding to keywords selected from the Internet from the viewpoint of Q3. Furthermore, the proposed system could provide suitable advice for the subject preferences from the viewpoint of Q4. Finally, Q5 denotes the proposed system that motivates subjects to exercise. The participants were highly evaluated from this viewpoint because they could passively and timely decide on exercise plans as the time they spent at home was tracked, and they received useful advice if they had stayed at home for too long.

Finally, subjective evaluations confirmed that, from the user experience perspective, the proposed system functioned correctly over three days, regardless of the diverse home environments where it was deployed. However, our subjective evaluation is a preliminary because the number of samples and the experiment duration are limited. Therefore, to obtain the quantitative results to reveal the performance of the proposed system from the viewpoint of promoting behavior change, further experiments with many subjects and various home environments are required.

## 5. Conclusions

In this study, an exercise support system based on an agent-oriented IoT architecture was proposed that enables people living alone to acquire a better daily lifestyle passively and continuously. The system advises users based on their situations and preferences.

From the perspective of system design, we modeled IoT devices as IoT agents playing simple roles and realized the autonomous cooperation among the distributed IoT agents based on an agent-oriented IoT architecture. This change brings flexibility to the system composition and deployment and the extensibility of implementing additional IoT agents without significant changes in the entire system. Furthermore, the proposed system is inherently cost-effective, utilizing affordable off-the-shelf IoT devices, sensors, and the user’s smartphone, without the need for specialized equipment. However, the actual implementation cost may vary based on home environments and the configuration of IoT devices.

From the perspective of service design, the proposed system realized exercise promotion using autonomous cooperation among the IoT agents. The proposed system was implemented in each subject’s home for the experiments. The proposed system offered advice by correctly detecting when the subject remained at home and recognizing the status. Experimental results proved that the proposed system can be implemented in certain home environments through the proposed system design. In addition, it adapts the advice content based on user preferences using feedback from the user within one or two weeks. Furthermore, the experiment also confirmed that people living alone benefit from the motivation to exercise when passively receiving advice that suits them based on subjective evaluations. Therefore, in conclusion, the proposed system can support daily life from the viewpoint of exercise to motivate participants to exercise and maintain a better lifestyle.

The proposed system is based on a decentralized approach using IoT agents, which distinguishes it from a cloud-based centralized approach. However, the comprehensive management of agents and services becomes necessary when deploying numerous agents and services due to the scalability challenges from a management perspective. Therefore, the implementation of an orchestration system for agents and services is necessary to leverage the benefits of both centralized and decentralized approaches.

While a major supervised learning algorithm was employed to estimate the user status, there is room for enhancing the accuracy of the status estimation. Additionally, the status estimation patterns can be diversified to improve the user experience and implement further services. In the future, various machine learning algorithms should be applied and evaluated for status estimation. Furthermore, the proposed system employed a simple JSON-based request-and-response messaging for agent communication. In contrast, various messaging designs, such as the REST API, MQTT [46], gRPC [47], and others, were considered. Therefore, there is potential to enhance the flexibility of the communication protocol design.

Furthermore, our experiments were preliminary, aiming to demonstrate the impact of the proposed system on behavior change However, due to the limited number of subjects and environments, large-scale and long-term experiments are necessary, involving sufficient subjects in various home environments to analyze the actual impacts on their behavior change and healthcare.

## Figures and Tables

**Figure 1 sensors-24-02029-f001:**
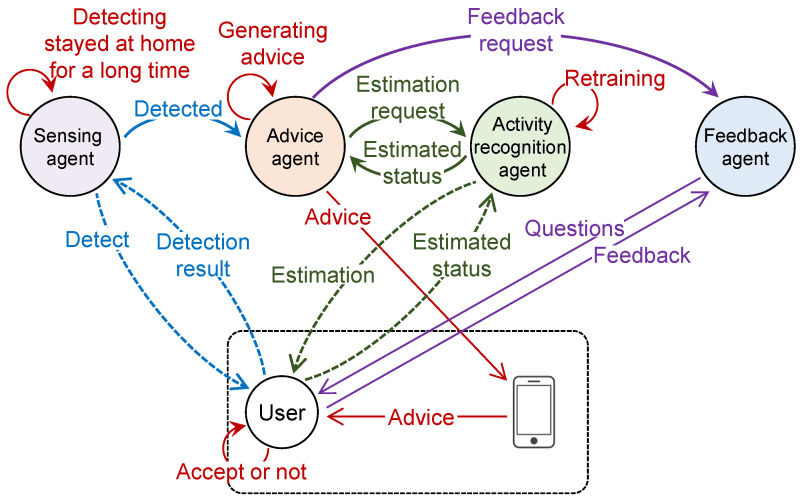
Relations and interactions among user, agents, and smartphone in the proposed system.

**Figure 2 sensors-24-02029-f002:**
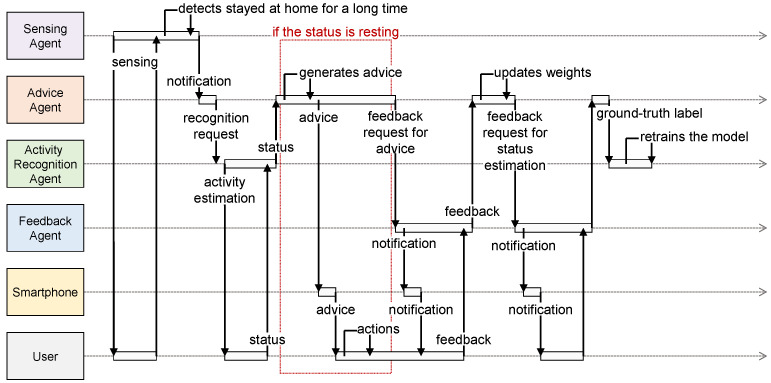
Example workflow of the proposed system when the sensing agent determines that the user has remained at home for an extended period.

**Figure 3 sensors-24-02029-f003:**
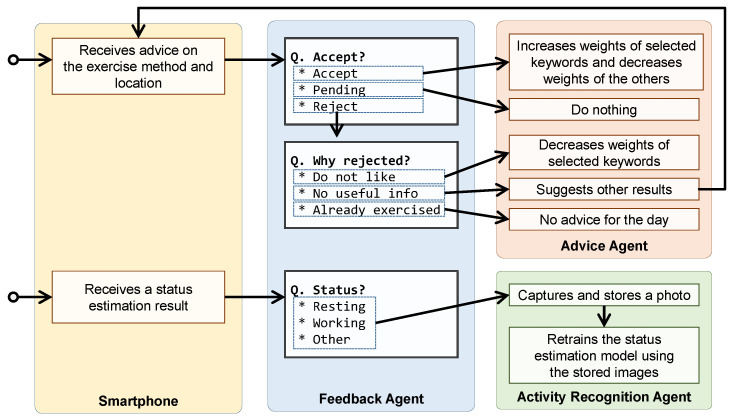
Feedback flow of the proposed system.

**Figure 4 sensors-24-02029-f004:**
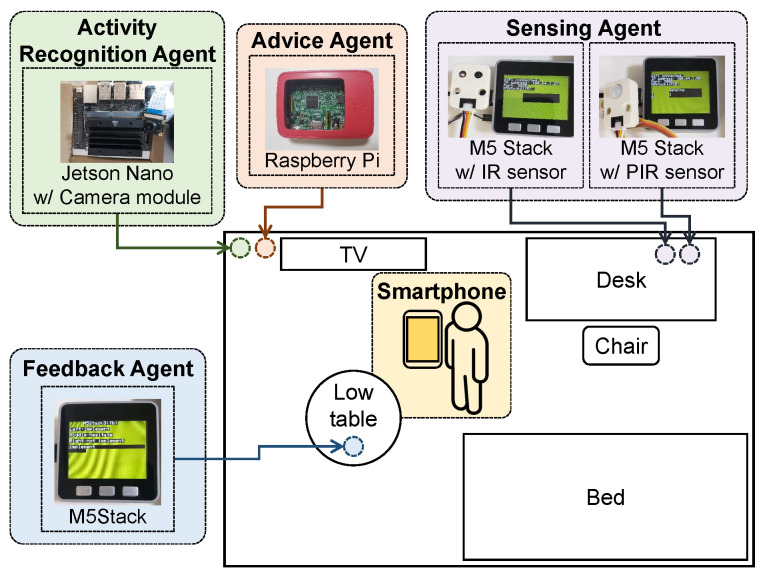
Actual deployment example in a subject’s home. The deployment layout depends on each subject’s homes.

**Figure 5 sensors-24-02029-f005:**
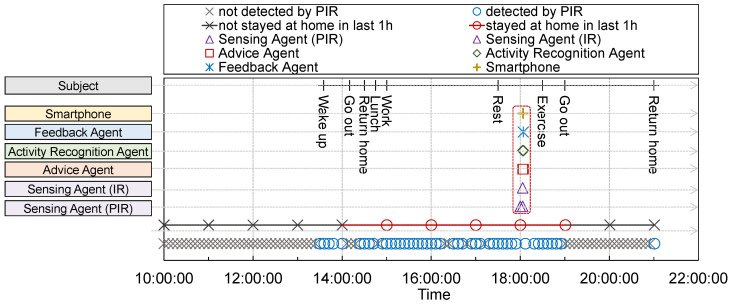
Time sequence of the proposed system in an experiment daily. The subject remained at home for a long time after going out and returning home; subsequently, the subject received advice at 18:03, when the system detected that the subject had been home for 4 h. A detailed cooperation sequence of agents from 18:03 to 18:04 is shown in Figure 6.

**Figure 6 sensors-24-02029-f006:**
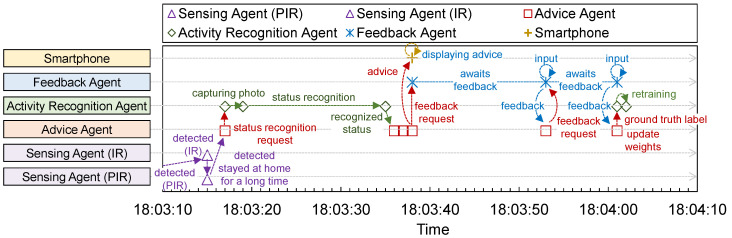
Time sequence of the proposed system focused on 18:03 to 18:04 in Figure 5. The advice agent received the notification when the sensing agents detected that the subject had been home for 4 h. Subsequently, they requested the status recognition of the activity recognition agent. The activity recognition agent captured a photo to recognize the status and returned the result. The advice agent generated and sent exercise advice to the subject’s smartphone and requested feedback from the subject using the feedback agent. Next, the subject returned two points of feedback, about the exercise and its recognized status. Subsequently, the activity recognition agent and advice agent updated the status recognition model and weights of the subject preference for exercise.

**Figure 7 sensors-24-02029-f007:**
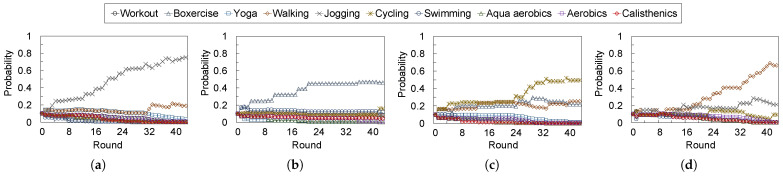
Transitions of the probabilities to suggestions for exercise of each subject. The probabilities adapt to the subject’s preferences of exercise types based on their feedback. Each suggestion probability is calculated by Equation (Equation 1) and the weight update algorithm of each keyword is described in Algorithm 4. (**a**) Subject A. (**b**) Subject B. (**c**) Subject C. (**d**) Subject D.

**Figure 8 sensors-24-02029-f008:**
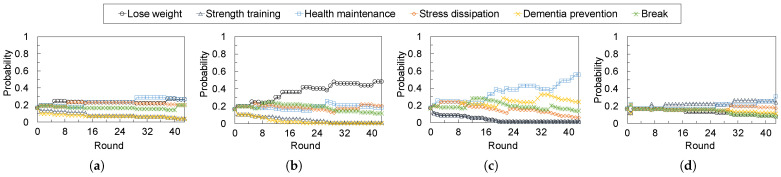
Transitions of the probabilities to suggestions for the exercise objective of each subject. The probabilities adapt to the subject’s preferences of exercise objectives based on their feedback. Each suggestion probability is calculated by Equation (Equation 1) and the weight update algorithm of each keyword is described in Algorithm 4. (**a**) Subject A. (**b**) Subject B. (**c**) Subject C. (**d**) Subject D.

**Table 1 sensors-24-02029-t001:** Subjective evaluation results of the proposed system after each subject experienced the proposed system for three days. The questions were translated from Japanese into English.

No.	Question	Score
Q1	Remaining at home for a long time was appropriately detected.	4.75
Q2	The subject’s status was appropriately estimated.	3.25
Q3	The suggested advice was appropriate for the selected keywords.	4.50
Q4	The suggested advice was appropriate for the subject’s preference.	4.75
Q5	This support system can motivate subjects to exercise.	4.50

## Data Availability

Data are contained within the article.

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
