# Peer review of "Exercise Promotion System for Single Households Based on Agent-Oriented IoT Architecture"

_sensors, 2024, doi:10.3390/s24072029_

Round 1

Reviewer 1 Report

Comments and Suggestions for Authors

I have reviewed the manuscript carefully entitle “Exercise Promotion System for Single Households Based on Agent-based IoT Architecture. My concerns are given below.

·      Title of manuscript should be revised; the word based twice looks awkward.

·      Abstract is too general and needs revision with pacific numeric information

·      Authors must introduce the set of exercises in the manuscript.

·      Figure 4 shows Jetson nano used as activity agent, while raspberry Pi used for advice agent, Why two different board/technology was used for activities, It would be better to used same board or module, Authors should present the depth prons, corns of these two for the said activity

·      On the table 2 what grounds the score were defined, need more detail

·      Figure 4 show only one part/room that contained bed, chair, table, desk TV, what about the larger home? Where more than one room and lawn.  

·      Figure 7 and 8 should be breakup in more figures and size of figures should be larger so that easily readable.

·      Every person does not have smart phone and such devices, authors should add cost analysis in the manuscript.

·      Manuscript is lack of focus discussion and authors need to add detailed discussion along with results.

·       Bench mark table should be added by giving the compression of current study with literature by giving its pros and cons with previous studies.

Comments on the Quality of English Language

need some improvement

Reviewer 2 Report

Comments and Suggestions for Authors

The purpose of this study was to present an exercise promotion system for single households. The proposed system is an agent-based IoT system and contains four agents and a smartphone. The system presented in the article is described methodically and the algorithms for each agent's operation are presented. This system was implemented in the homes of four subjects and was tested for three days. The test result is provided in the article.

The article does not provide a sufficient justification for what makes the system unique and does not emphasize the originality of the presented work compared to the work of other authors. In the section on related work, it is generally mentioned that there are similar systems. However, the authors do not reveal which of the systems are the most similar and how the proposed system will be superior compared to already developed systems and solutions or how the proposed system will be useful for creating new IoT systems. The analysis of related work does not reveal the uniqueness of the presented work. The originality and novelty of the work should be highlighted more.

The result and discussion section is provided, but there are no discussions in this section. The authors should discuss the results and how they can be interpreted from the perspective of previous studies and working hypotheses. The findings and their implications should be discussed in the broadest possible context and the limitations of the work should be highlighted.
The system has been tested in 3 user homes, but after these system tests, there are no conclusions about what should be improved or paid more attention to in case of further development of the system.

The created system is characterized as a system where agents work autonomously in it. However, the protocols, message formats, and APIs used for the communication of these agents are not provided.

It is not clear what the extensibility of the developed systems would be. Is it possible to implement more new agent devices in this system?

The system was tested in one-room houses. What modification is needed if the house has more than 1 room and more sensing and activity recognition agents must be installed?

The probabilities of suggestions for exercise and exercise objective are provided in Figures 7 and 8, but there are no explanations why each figure has four plots. The authors do not explain how these probabilities were calculated.

Reviewer 3 Report

Comments and Suggestions for Authors
  • The description of related research is too scant. More details on the relevant studies are needed to facilitate a comparison between the related research and the results of the current study.

  • Experimental Results: The experiment was conducted with only 4 participants, which seems insufficient. Particularly in qualitative assessments such as surveys, targeting only 4 individuals may be excessively limited.

  • There are likely pros and cons to strong and weak integration with the system. Please elaborate further on this, providing a more detailed explanation.

  • The conclusion section appears somewhat weak. Provide a more comprehensive discussion of the study's results, strengths, weaknesses, and areas for improvement.

Comments on the Quality of English Language

The authors have to get some English proof. 

(lyfestyle is correct?)

Round 2

Reviewer 1 Report

Comments and Suggestions for Authors

The response is satisfactory therefore I recommend it for publication 

Comments on the Quality of English Language

Ok